# Body-Size Perception among First-Generation Chinese Migrants in Italy

**DOI:** 10.3390/ijerph19106063

**Published:** 2022-05-17

**Authors:** Giovanni Castellini, Alessio Pellegrino, Livio Tarchi, Maria Calabrese, Maria Boddi, Valdo Ricca, Gianfranco Costanzo, Pietro Amedeo Modesti

**Affiliations:** 1Department of Neuroscience, Psychology, Drug Research and Child Health, University of Florence, 50121 Florence, Italy; giovanni.castellini@unifi.it (G.C.); livio.tarchi@unifi.it (L.T.); valdo.ricca@unifi.it (V.R.); 2Department of Experimental and Clinical Medicine, University of Florence, 50121 Florence, Italy; maria.boddi@unifi.it (M.B.); pa.modesti@unifi.it (P.A.M.); 3Diabetology Unit, Ospedale Misericordia e Dolce, 59100 Prato, Italy; maria1.calabrese@uslcentro.toscana.it; 4National Institute for Health, Migration and Poverty, 00153 Rome, Italy; gianfranco.costanzo@inmp.it

**Keywords:** cardiovascular prevention, overweight or obesity, ethnicity, health policies, minority groups, immigration, migrant

## Abstract

Body-size perception is an important factor in motivating people to lose weight. Study aim was to explore the perception of body image among first-generation Chinese migrants living in Italy. A sample of 1258 Chinese first-generation immigrants and of 285 native Italians living in Prato, Italy, underwent blood pressure measurements, blood tests (with measurement of glucose, cholesterol, and triglycerides), and anthropometric measurements. Body-size perception was investigated with Pulvers’ figure rating scale using logistic or linear multivariable regression adjusted for age, gender, BMI, education and years spent in Italy. Chinese migrants had lower BMI and discrepancy score (preferred minus current body size) than Italians (*p* < 0.05 for both). After a logistic regression analysis, the discrepancy score remained lower in the Chinese than in the Italian cohort independently from BMI and other confounders (OR 0.68; 95%CI 0.50 to 0.92). In the Chinese cohort, female gender, BMI and years spent in Italy were positive determinants of discrepancy score (desire to be thinner), while age showed negative impact (*p* < 0.05 for all). Overweight is an important risk factor for diabetes, a very prevalent condition among first-generation Chinese migrants. The present study offers useful information and suggests the need for prevention programs specifically addressed to men.

## 1. Introduction

The prevalence of obesity has progressively increased over the past two decades at a global level and, in most of the world, overweight is now becoming the new “normal” weight [1]. It is commonly stated that urbanization is one of the most important drivers of the worldwide rise in body mass index (BMI), because diet and lifestyle in cities lead to adiposity [2,3,4]. More recent findings contradict this dominant paradigm, because more than 55% of the global rise in mean BMI from 1985 to 2017, and more than 80% in some low- and middle-income regions, was found to be due to increases in BMI in rural areas [5]. Having a low level of education and cultural factors may limit the receipt of prevention messages by people living in rural areas [6,7,8]. Failure to recognize the health risks associated with obesity could perpetuate a false sense of security about well-being and perpetuate unhealthy lifestyles [9]. Body image perceptions may modulate actions related to weight loss and weight control; a certain degree of dissatisfaction towards one’s own current body image was reported to be an important factor in motivating healthy habits in the general population [10,11]. The relationship between body image and actual weight status was reported to be influenced by gender [12], socioeconomic status [13], countries and cultures [14,15,16]. Caucasian women were reported to experience body dissatisfaction at lower BMI levels than black women, whereas black women were reported to perceive themselves as being a normal weight when they are overweight [14,15]. According to the International Health and Behavior Survey, being female was related to feeling overweight at any BMI decile [14], a perception especially high in Asian women [14]. In Japan, similar to in Korea, underweight women view themselves as normal or overweight [17,18], and a high percentage of normal-weight women believed themselves to be overweight [19]. However, body image perception in Asians is a complex issue because of the lower BMI cutoff for overweight/obesity established for Asian in comparison to Caucasians populations [20]. Nowadays, overweight/obesity control in Asian countries is crucial because of the close relationship with diabetes, both in these areas of the world and among Asian migrants living in Western countries [21,22,23,24,25,26]. However, until now, no study has investigated the interaction between body image perception and BMI in Chinese migrants living in Europe. Therefore, the present study was designed to investigate (1) the potential difference between Chinese first-generation migrants and native Italians for what concerns desired body shape, and the distance between the currently perceived and the desired body shape; (2) the interaction between body image perception and ethnically specific BMI standards; and finally, (3) the relationship between body-size perception and time spent in Italy since migration, an indirect measure of acculturation.

## 2. Materials and Methods

### 2.1. Setting and Study Population

The present study was performed within the frame of the CHinese In Prato project (CHIP), a participatory research project designed to investigate the health needs of the Chinese community resident in Prato, Italy [22]. Located in Tuscany, 30 km from Florence, Prato has a population of approximately 200,000 inhabitants and one of the highest proportions of Chinese immigrants in any Italian province, as well as one of the highest in Europe [23,24,25]. A community–academic partnership composed of the Consulate General of Florence, local community-based Chinese organizations, and the Chinese and Italian Universities, was created to lead the CHIP project [23,24,25]. In the CHIP project, the Chinese population was enrolled in a cardiovascular risk-factor screening program through a sensitive, culturally appropriate, non-coercive recruitment process that adopted a network sampling procedure, as previously described [23,24,25,26]. Each participant received the results of all performed clinical and biochemical tests, with a clear statement of whether the diagnostic criteria for hypertension, type 2 diabetes, or dyslipidemia were met. Participants with screen-detected diseases were then offered second-level assessments and treatments through the Regional Health System, with clinical practices based on current international guidelines. Data were collected between July 2014 and November 2019; they were anonymous and de-identified. A contrast with the native Italian population was obtained through a cohort enrolled in 2014 and randomly sampled from General Practice lists stratified by age and gender. Each control subject was initially sent a letter informing them about the study, followed by an invitation to attend a screening. The subjects were replaced after two invitations. Those with whom no contact was established after three invitations were sent a letter by recorded delivery mail. The response rate of the eligible Italian subjects approached during recruitment was 67%. The exclusion criteria included pregnant women, critically ill individuals, and impaired cognitive ability, as judged by clinical staff members during the first evaluation. No subjects were excluded. The study was approved by the Ethical committee of the Azienda Ospedaliero-Universitaria Careggi (Ref. OSS.14.089). Written informed consent was obtained from all participants. The subjects were provided with a written description of the study in their choice of Chinese or Italian language, and written consent was obtained at the time of entry from each participant. Participants with untreated clinical diseases identified during the examinations were advised to see their general practitioner or referred to the Hospital of Prato. No other incentives were offered to study participants.

### 2.2. Data Collection

All participants were instructed to fast overnight before the day of survey. In the early morning (between 07.00 and 10.00 a.m.), individuals attended the Research Centre, where trained Chinese and Italian staff members administered a questionnaire and performed physical (blood pressure and anthropometry) and biochemical blood measurements (glucose, total cholesterol, and triglycerides). The gathered information included participant sociodemographic data, tobacco use, alcohol consumption, medical and migration history. The Pulvers’ figure rating scale [27] was also administered. Body weight, height, and waist and hip circumferences were then measured according to standardized protocols, and body mass index (BMI) was calculated as previously reported [23,24,25]. Blood pressure (BP) was measured 3 times according to current guidelines [28] using a clinically validated semiautomatic digital sphygmomanometer (M6; Omron Matsusaka Co., Ltd., Matsusaka, Japan) [26]. The average of the last two readings was used for analysis. Biochemical measurements were finally performed on fingerpick blood samples using validated dry chemistry methods (AccuChek AVIVA, Roche Diagnostics S.p.A., Mannheim, Germany for glucose and MultiCare-in, HPS, Italy, for total cholesterol and triglycerides) [23,24,25]. Non-fasting participants were asked to return at fast for blood tests. Participants with fasting glucose ≥ 126 mg/dL, systolic BP ≥ 140 mmHg, or diastolic BP ≥ 90 mmHg were also asked to return for confirmatory testing. All requested participants attended the second visit.

### 2.3. Diagnostic Criteria

Hypertension was diagnosed if systolic BP (SBP) was ≥140 mmHg, or diastolic BP (DBP) was ≥90 mmHg at the two visits, or if anti-hypertension medication had been taken in the previous two weeks [28]. The diagnosis of type 2 diabetes was based on fasting plasma glucose criteria (≥126 mg/dL confirmed by repeated testing at a second visit) and/or current treatment with glucose-lowering drugs. Type 1 diabetes was defined by clinical parameters, including absolute need for insulin, young age of onset, and history of ketosis, for the purpose of this study. High cholesterol was classified for total cholesterol (TC) levels ≥ 240 mg/dL [28], and high triglycerides (TG) was classified for TG ≥ 200 mg/dL [28]. Other categorized exposures were education level (no studies, primary and secondary school, high school, college or more), alcohol use, current smoking (noncurrent smokers defined as those who had never smoked and former smokers who-had stopped smoking), health insurance (none; STP card, Foreigner Temporary Present; full registration to the Regional Health Service), ability to speak the Italian language (yes or no), and migration history (years of residence in Italy, area of origin in China). Silhouettes were categorized with a progressively larger body size. A discrepancy score was created by subtracting the number of the silhouette selected as the preferred body size from the number of the silhouette selected as the current body size. Therefore, the discrepancy score can be seen as a measure of the desire to be smaller, a high value indicating a low satisfaction with body size.

### 2.4. Statistical Analysis

For descriptive statistics, means and standard deviations were given for each sample. When variables were dichotomous or categorical, percentages were offered. Group differences were estimated using χ^2^ tests for categorical variables, or t-tests for continuous variables. Comparisons between Chinese and Italian cohorts were made via logistic regression analysis (the appropriate regression analysis to conduct when the dependent variable is dichotomous); this was also adjusted for age, gender, BMI, marital status and education level. Differences in discrepancy score between the two cohorts, overall participants and by gender, were investigated using a multiple logistic regression analysis that was also adjusted for age, sex, BMI (as mean value or through dummy categorical variables according to the ethnic-specific cut points) and education level. The determinants of discrepancy scores in the Chinese cohort were investigated at multivariate linear regression analysis using two different regression models (Model 1 including age, gender, BMI, education level, urban or rural area of origin in China, alcohol use, smoking habit, health perception; Model 2 including also years spent in Italy). The regression coefficients (B) with 95% CL were reported. IBM SPSS software (version 27.0, SPSS Inc., Chicago, IL, USA) was used for analysis.

## 3. Results

### 3.1. Characteristics of Participants

Overall, 1258 Chinese first-generation migrants and 285 Italians were enrolled. The characteristics of the study participants stratified by country of birth are presented in Table 1.

Chinese migrants were significantly younger, more frequently married, and had lower educational level than the Italian participants. In the Chinese migrant cohort, 189 subjects did not complete primary education, 856 (69%) were not able to speak Italian, and 734 (58%) had no free access to healthcare.

The main characteristics of the Chinese and Italian participants are reported in Table 1. Chinese migrants were significantly shorter and lighter and had lower BMIs than the Italian participants. Waist and hip circumferences and the waist-to-hip ratio were also significantly smaller in Chinese participants. In the logistic regression analysis (adjusted for age, marital status, and educational level), Chinese migrants had a lower BMI than native Italians across both men (OR 0.81; 95% Cl 0.75 to 0.89) and women (0.89; 95% Cl 0.83 to 0.95).

### 3.2. Body Image Perception

For the analysis, silhouettes were categorized with a progressively larger body size. Chinese migrants chose significantly smaller current (“see you”) and preferred (“would be”) silhouettes than the referent Italians for both genders included in the study (*p* < 0.001, for all χ^2^ tests). Overall, Chinese migrants had a lower discrepancy score (a measure of the desire to be smaller) than Italians (0.55 ± 1.01 and 0.91 ± 1.01, respectively, *p* < 0.05). The distribution of discrepancy score by categories (negative value, 0, positive value) and gender in Chinese and Italians is shown in Figure 1.

Importantly, in the logistic regression analysis (adjusted for age, sex, BMI, and education level), the discrepancy score category remained lower among Chinese migrants than among Italians, independently from BMI (OR 0.68; 95% Cl 0.50 to 0.92). Most importantly, the different body image perception between Chinese and Italians also persisted when BMI was classified using ethnically specific categories (Table 2), with the Chinese participants consistently showing a lower desire to be thinner than Italians.

When logistic regression analysis was performed by gender using the same model, Chinese women showed a markedly lower discrepancy score than Italian women (OR 0.28; 95% Cl 0.17 to 0.46, *p* < 0.001), whereas no differences were detectable between Chinese and Italian men (OR 0.89; 95% Cl 0.62 to 1.29, *p* = 0.550). Women showed higher desire to be thinner than men because the OR of discrepancy score (adjusted for age, BMI, and education) was 3.01 (95% Cl 2.56 to 3.70) among Chinese and 1.62 (95% Cl 1.23 to 2.15) among Italians. The discrepancy score in the Chinese cohort by gender and BMI ethnically specific categories is shown in Figure 2.

Determinants of discrepancy score in the Chinese cohort were finally investigated at multivariate linear regression analysis adjusted for age, gender, BMI, education level, urban or rural area of origin in China, alcohol use, smoking habit, health perception, and years spent in Italy. The desire to be thinner (discrepancy score) of Chinese migrants was positively influenced by female gender, BMI and years spent in Italy, whereas age showed a negative impact (Table 3).

## 4. Discussion

To the best of our knowledge, this is the first study which evaluates the body image perceptions of first-generation Chinese migrants living in Europe. The main results, and the clear message for future prevention programs, are that (1) Chinese migrants had a lower desire to be thinner than Italians; (2) this different perception was independent from the lower BMI recorded among Chinese than among Italians; and finally, (3) positive determinants of the desire to be thinner in the Chinese cohort were gender (women generally a higher discrepancy score) and time (measured in years—those in Italy for longer generally had a higher discrepancy score) spent in Italy.

Chinese migrants chose significantly smaller preferred silhouettes less often than the referent Italians. This drive for thinness was represented not only in desired body shapes, but also in the subjective evaluations of one’s own status. Indeed, at the moment of evaluation, Chinese migrants were also more likely to represent themselves thinner than Italians, supporting the hypothesis of a reduced drive for thinness in Chinese people. The discrepancy score used in this study is only one of the possible indicators of subjective perception of body size. Further studies using different indicators of the desired body size are likely to reveal more specific and accurate results. The lower BMI of Chinese migrants than Italians might have been responsible for the difference. The WHO consultation group recommended a lower cut-off of BMI for Asian people with respect to European populations [20], and the identified diagnostic cut-off for overweight in Chinese people was 24 kg/m^2^ [29]. However, in the present study, differences in body perception were independent from ethnically specific BMI stratification. An ethnically specific diagnostic cut-off was indeed considered at multivariate regressions, signaling a higher contribution of subjective factors in the appraisal of one’s own body shape. This result is in line with the previous literature on the topic of embodiment—that is, the lived experience of individuals in relation to their body, their bodily function, and their bodily stimuli. These lived experiences of the individuals, in respect to their body, the embodiment, has previously been linked to both self-esteem and disordered eating, with a stark contrast between genders.

In concordance with the previous literature, in the current study, gender significantly influenced the desire for thinness because the discrepancy score (adjusted for age, BMI, and education) was 3 and 1.6 times higher in women than in men both in the Chinese and the Italian cohort, respectively. The present results also corroborate the findings obtained in other Asian or Chinese populations [14,19], where perceived weight was previously observed to be higher among women than men at any BMI decile. Chinese migrants reported a lower discrepancy score between perceived and desired body shapes in comparison to Italians; Chinese migrants also expressed a greater satisfaction with their body size in general. However, only Chinese women had less desire to be thinner than Italians (OR 0.28). On the contrary, Chinese men showed a similar desire for thinness than Italians (OR 0.89). According to the current study, women of Chinese descent exhibit a lower level of apprehension for body shape and weight concern in comparison to Italian women.

Nonetheless, one of the main findings of the current study, is the observation that the desire for thinness in Chinese migrant women who have relocated to Italy increased proportionally with time spent in the country. This specific aspect, which is a proxy of acculturation, has been severely understudied in the current literature, despite being known to significantly influence the subjective experience of individuals in respect to satisfaction with their body and their desire for thinness. Qualitative studies have found that females tend to desire weight loss to achieve the thin ideal portrayed by celebrities in the US [12]. The various body types and shapes portrayed by the media in television programs, films and magazines can shape the viewers’ perception of “standard” weight, as well as the perception of overweight or obese individuals [10,30]. Specifically, it is considered that overweight or obese people are generally stigmatized, and the media express a specific preference for thinness [30,31]. In order to judge a factor as noxious, or casually linked to an outcome, several criteria have been proposed, and little or no agreement is generally reached on how to define the overarching indicators. However, the criteria of exposure to the noxa, temporal precedence, dose-dependency and plausibility have been more commonly proposed and more widely appraised. Acculturation seems to satisfy all factors defined as causally linked to the desire for thinness because, with all else being equal (e.g., biological or ethnic factors), the desire for thinness was observed to be conditioned by social and cultural factors. Specifically, exposure to cultural and societal beauty standards, posited to be gender-specific, would be expected to be more significant for women. A wide range of factors can mediate the difference between functional and dysfunctional weight control [32], but cultural elements are particularly relevant for public health considerations [33,34]. The exposure to beauty standards precedes the desire for thinness, which satisfies the criterion for temporal precedence. The more time spent in the country, the more desire to adhere to the cultural standard would be observed, according to the notion of dose-dependency and plausibility, which was also satisfied. Age was observed to be negatively associated with desire for thinness in the population of first-generation Chinese migrants. As reported above, in both the Italian and the Chinese cohort, the discrepancy score was positively influenced by gender. Men tended to express a lower desire for thinness than women, which may predispose them to a higher burden of cardiovascular or metabolic risks. Health-promotion strategies and weight-control programs targeting men, irrespective of ethnicity, seem warranted. The implications for ethnic or gender studies on health problems are relevant in the development of new prevention strategies targeting the migrant population. Metabolic diseases are a major problem among Chinese migrants in Europe [21,22,23,24,25,26]. However, linguistic or cultural difficulties can make it difficult to convey this message to the migrant population, because messages are generally intended for the resident population. This point is also very important considering the different characteristics of Chinese migrants compared to the native resident population. Indeed, Chinese migrants are not only significantly younger and more frequently married, but also have a lower level of education than Italian participants. This strongly characterizes the migrant population enrolled in this empirical study compared to other populations enrolled in previous Asian studies such as Japan and Korea [17,18]. The direct participation of the Chinese community in a common project is to be acknowledged.

The limitations of the current study also have to be considered. Snowball sampling may hinder the generalization of the current results, as other sampling strategies may optimize a proper representation of the underlying characteristics of a reference population. However, undocumented migrants are usually excluded by conventional approaches, whereas snowball sampling guarantees adequate response rates in high-risk groups. Second, different sampling methods might also affect results. More precisely, Chinese participants with a greater interest in their own weight and size would be more likely to participate in the survey. Therefore, at most, the observed differences could be underestimated. Third, the cross-sectional design of the study does not permit a direct appraisal of the direction of causality between the collected variables and acculturation process. However, the study has also clear advantages because all anthropometric variables were clinically measured rather than self-reported [35].

## 5. Conclusions

In conclusion, the present study offers preliminary evidence of differences in body shape self-appraisal in Chinese migrants compared to Italians, with significant factors being gender and acculturation. More precisely, Chinese migrants had a lower desire to be thinner than Italians; the differences between country of origin were independent from BMI; and finally, gender and years spent in Italy were important determinants of the desire to be thinner in the Chinese cohort. In Italy, more attention is paid to healthy food than to physical activity, especially among women [36]. Chinese migrant women were observed to change their diet quality as they spent more years in Italy, achieving a reduction in body weight and blood cholesterol [37]. Specifically, adherence to the Mediterranean Diet score increased with the length of residence in Italy only among women [37]. Migrant women appear to be more receptive than men to cardiovascular-risk-prevention messages. Programs aimed at women are needed to support the acculturation process of men.

## Figures and Tables

**Figure 1 ijerph-19-06063-f001:**
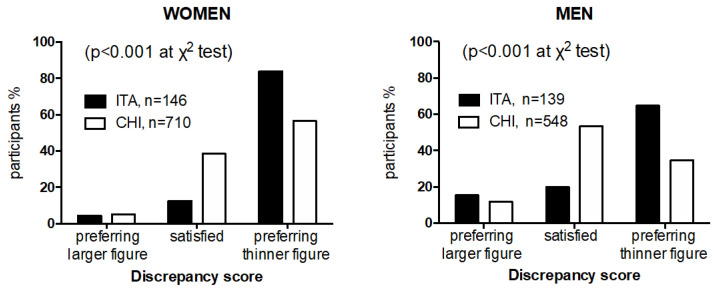
Distribution of discrepancy score categories (negative value, 0, positive value) in Chinese migrants and native Italians by gender.

**Figure 2 ijerph-19-06063-f002:**
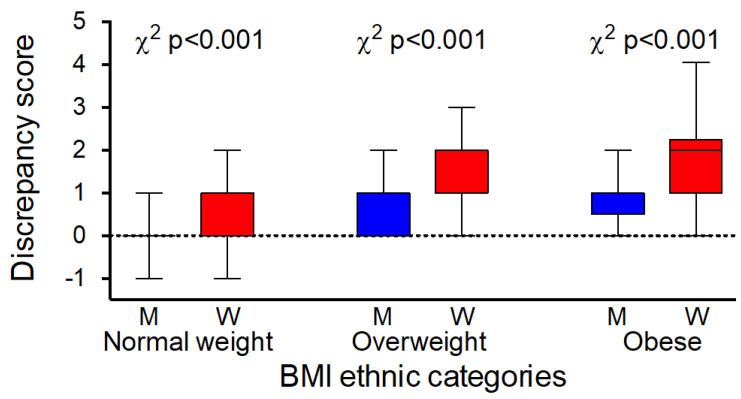
Distribution (length of the box is the difference between the 75th and 25th percentiles, ends of the box are 9-95th percentile) of discrepancy score in the Chinese cohort by gender (M, men and W, women) and BMI ethnically specific categories.

**Table 1 ijerph-19-06063-t001:** Sociodemographic and clinical characteristics of the Italian (*n* = 285) and Chinese (*n* = 1258) participants.

	Italians	Chinese	*p*
**Categorical Variables, n (%)**			
Age decades			<0.001
*30–39 years*	63 (22.1)	399 (31.7)	
*40–49 years*	108 (37.9)	496 (39.4)	
*50–59 years*	114 (40.0)	363 (28.9)	
Women	146 (51.2)	710 (56.4)	0.010
Health insurance			–
*RHS*	285 (100.0)	294 (23.4)	
*TPF*	–	226 (18.0)	
*No*	–	734 (58.3)	
Education			<0.001
*Illiterate*	–	189 (15.0)	
*Primary school*	36 (12.6)	556 (44.2)	
*Middle school*	186 (65.3)	487 (38.7)	
*College or more*	63 (22.1)	25 (2.0)	
BMI ethnically specific categories			<0.001
*Normal*	126 (44.2)	705 (56.0)	
*Overweight*	138 (48.4)	430 (34.2)	
*Obese*	21 (7.4)	123 (9.8)	
Smokers	48 (16.8)	235 (18.7)	0.632
Alcohol use	189 (66.3)	569 (45.2)	<0.001
Hypertension	58 (20.4)	266 (21.1)	0.832
Diabetes	21 (7.4)	157 (12.5)	0.015
**Continuous variables, mean ± SD**			
Age (years)	47.5 ± 7.4	44.3 ± 8.1	<0.001
Height (cm)	171.2 ± 10.8	163.2 ± 7.8	<0.001
Weight (kg)	74.6 ± 12.1	63.5 ± 10.7	<0.001
BMI (kg/m^2^)	25.4 ± 3.2	23.8 ± 3.1	<0.001
Waist (cm)	89.2 ± 9.8	82.5 ± 9.6	<0.001
Hip (cm)	99.3 ± 6.8	95.5 ± 6.4	<0.001
Waist/Hip ratio	0.898 ± 0.077	0.863 ± 0.070	<0.001
SBP (mmHg)	120.4 ± 13.9	119.6 ± 19.0	0.107
DBP (mmHg)	77.7 ± 10.7	79.9 ± 11.7	<0.001
Glucose (mg/dL)	103.5 ± 12.9	116.8 ± 31.2	<0.001
Cholesterol (mg/dL)	190.6 ± 55.6	235.2 ± 64.2	<0.001
Triglycerides (mg/dL)	163.0 ± 87.2	196.8 ± 107.2	<0.001

RHS = full registration to Regional Health System; TPF = registration as Temporary Present Foreigner; BMI = body mass index; SBP = Systolic Blood Pressure; DBP = Diastolic Blood Pressure.

**Table 2 ijerph-19-06063-t002:** Differences between Chinese migrants and native Italians at multivariate logistic regression (adjusted for all exposures reported in the table).

Exposures	OR (95% Cl)	*p*
Age (years)	0.921 (0.903 to 0.939)	<0.001
Gender (women)	0.974 (0.702 to 1.350)	0.873
BMI (ethnically specific categories)	0.824 (0.588 to 1.154)	0.260
Education (categories)	0.135 (0.102 to 0.179)	<0.001
Discrepancy Score (thinner figure)	0.546 (0.409 to 0.730)	<0.001

BMI = body mass index.

**Table 3 ijerph-19-06063-t003:** Determinants of discrepancy score (categories) in the Chinese migrant cohort at multivariate linear regression.

Exposures	Model 1		Model 2	
	**B (95% Cl for B)**	** *p* **	**B (95% Cl for B)**	** *p* **
Gender (women)	0.386 (0.313 to 0.459)	<0.001	0.425 (0.319 to 0.531)	<0.001
BMI (kg/m^2^)	0.102 (0.092 to 0.112)	<0.001	0.101 (0.087 to 0.115)	<0.001
Age (years)	−0.008 (−0.012 to −0.004)	<0.001	−0.009 (−0.015 to −0.003)	0.002
Time in Italy (years)	-	-	0.009 (0.000 to 0.018)	0.042
*Multiple r=*	*0.553*		*0.557*	

BMI = body mass index. Model 1 adjusted for age, gender, BMI, education level, urban or rural area of origin in China, alcohol use, smoking habit, health perception (n = 1241 included in analysis); Model 2 adjusted also for years spent in Italy (n = 618 included in analysis).

## Data Availability

Data are available from the CHIP study. P.A.M. is the Scientific Coordinator of CHIP and may be contacted with further questions.

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
