# Peer review of "Body-Size Perception among First-Generation Chinese Migrants in Italy"

_ijerph, 2022, doi:10.3390/ijerph19106063_

Round 1

Reviewer 1 Report

This research topic is interesting and important. At the current stage, the research has elaborated on the data collection and analysis. More work is needed to highlight the importance of this research in terms of migrants’ health issues or relevant health policies.

The following are suggestions to beef up this research.

First, what is logistic regression analysis? Please specify the method, and Model 1, and Model 2 in Table 3 in your methodology part (e.g., what is B, and what is B’s meaning)?

Second, one of the limitations is snowball sampling (Line 284). But the sampling for the native people is more random. Will this non-random sampling method exaggerate or underestimate the results? For instance, people who are more concerned with their weight and body size would be more likely to participate in the survey.

Third, in the conclusion part, the authors can refer to more implications from “migratory features”. For instance, Chinese migrants are “significantly younger, more frequently married, and had lower educational level than the Italian participants”. “Chinese migrants were significantly shorter and lighter and had lower BMIs than the Italian participants.” What is the implication for the ethnic or gendered studies on health issues? Please specify the value of this empirical study to the wider health or migration studies. (At least, a comparative conclusion with the prior studies on Asia such as Japan and Korea is needed.)

Fourth, Prato has a population of approximately 200,000 inhabitants and one of the highest proportions of Chinese immigrants in any Italian province, as well as one of the highest in Europe. Why can this city in Italy attract so many Chinese? Is it different from Paris or London where some Chinese migrate and congregate?

Fifth, the conclusion can also refer more to the cultural and social meaning of why are the years spent in Italy important determinants of the desire to be thinner among the Chinese migrants. 

Reviewer 2 Report

There is a growing body of literature elucidating the link between subjective weight perception and motivation to lose weight. Using a unique sample of 1,258 first-generation immigrants from China and 285 native-born Italians residing in Prato, Italy, the present study explores the perception of body image and the desire to be thinner. The desire to be thinner was measured using a discrepancy score between the preferred body size and the current body size. Consequently, the discrepancy score was interpreted by the authors as a measure of the desire to be smaller, with a higher value indicating lower satisfaction with one’s body size. The research sample was recruited at an undisclosed research center, where participants underwent blood pressure measurements, blood tests examining basic blood glucose levels and lipid profiles, and anthropometric measurements. Body size perception was investigated with the Pulvers’ figure rating scale (Pulvers et al., 2013). Analytic predictors included age, gender, health insurance coverage, BMI, educational attainment and years spent in Italy, among a few others. The logistic regression analyses revealed that Chinese migrants had lower BMI and discrepancy scores than Italians. Among the Chinese immigrants, being a female, being younger, having a higher BMI, and years spent in Italy were found to be important predictors of the desire to be thinner.

Overall, the present study offers useful insights into the discrepancies by ethnicity and immigrant status between the projected (desired) and real body sizes in a sample of otherwise healthy adults in Italy. The conclusions that are drawn from this study emphasize the need for preventive interventions aimed at vulnerable groups, specifically targeting men. The manuscript is generally accurately written, no typographic errors have been noticed. However, this study is not without limitations. Specifically, the study relied on non-random, accidental sampling that limits the overall generalizability of the results. It should also be noted that a discrepancy score used in this study is only one of the possible indicators of subjective perception of body size. Further studies that use different indicators of the desired body size are likely to reveal more specific and accurate results.

Round 2

Reviewer 2 Report

The authors have incorporated comments and constructive criticism from the referee(s) and the editor(s), which led to a better manuscript.